# Bacteriophage T5 tail tube structure suggests a trigger mechanism for *Siphoviridae* DNA ejection

Charles-Adrien Arnaud[1], Grégory Effantin[1], Corinne Vivès[1], Sylvain Engilberge[1], Maria Bacia[1], Pascale Boulanger [2], Eric Girard [1], Guy Schoehn [1] & Cécile Breyton [1]

The vast majority of phages, bacterial viruses, possess a tail ensuring host recognition, cell wall perforation and safe viral DNA transfer from the capsid to the host cytoplasm. Long flexible tails are formed from the tail tube protein (TTP) polymerised as hexameric rings around and stacked along the tape measure protein (TMP). Here, we report the crystal structure of T5 TTP pb6 at 2.2 Å resolution. Pb6 is unusual in forming a trimeric ring, although structure analysis reveals homology with all classical TTPs and related tube proteins of bacterial puncturing devices (type VI secretion system and R-pyocin). Structures of T5 tail tubes before and after interaction with the host receptor were determined by cryo-electron microscopy at 6 Å resolution. Comparison of these two structures reveals that host-binding information is not propagated to the capsid through conformational changes in the tail tube, suggesting a role of the TMP in this information transduction process.

[1] Université Grenoble Alpes, CNRS, CEA, Institut for Structural Biology (IBS), 38000 Grenoble, France. [2] Institute for Integrative Biology of the Cell (I2BC), CEA, CNRS, Université Paris-Sud, Université Paris-Saclay, 91405 Gif-sur-Yvette, France. Correspondence and requests for materials should be addressed to C.B. (email: Cecile.Breyton@ibs.fr)

Phage tail architectures and strategies of cell wall recognition and perforation are different for each family of tailed phages: *Myoviridae* use a 'syringe-like' mechanism, whereby the long and straight contractile tail mechanically and chemically 'drills' the cell wall with a metal-loaded needle[1–4]. For the short tailed *Podoviridae*, the sequence of events has also been investigated and proteins involved in DNA delivery have been identified: following attachment of the receptor fibres to the host, conformational changes are transmitted to the core proteins located in the capsid, leading to their expulsion, resulting in the formation of a channel that spans the whole cell wall[5–8]. Concerning the large family of the *Siphoviridae*, structural information (refs. [9, 10] and references therein) has yet to detail the mechanisms by which receptor binding promotes DNA ejection. A study based on negative-stain electron microscopy (EM) data proposed the tail tube protein (TTP) to be involved in signal transduction from the distal end of the tail to the capsid in *Bacillus subtilis* phage SPP1[11]. Beyond differences between the phage families, the wealth of structural data on phage tail proteins points to strong structural homologies, highlighting a common protein building block that has been duplicated and decorated with different domains to serve alternative functions within the long phage tails (e.g. refs. [9, 12]). Furthermore, structural homologies between tail proteins, proteins of the type VI secretion system (T6SS) of pathogenic bacteria[13, 14] and of R-pyocins[15] suggest a common evolutionary origin that evaded sequence analysis because of very low sequence conservation (reviewed in refs. [9, 14]). In particular, the inner tube of all these puncturing devices is formed by the stack of doughnut-shaped, structurally very conserved hexamers.

Phage T5 is a *Siphoviridae* infecting the Gram-negative host *Escherichia coli*. Its overall structure has been determined[16], showing an unusual threefold symmetry of the tail tube[17]. To our knowledge, such a threefold symmetric tail tube has been observed in only one other case, for siphophage ΦCbK[18, 19]. In T5, the tail tube is formed by the stack of 40 trimers of the TTP pb6[16] around the tape measure protein (TMP). The latter spans the whole tail tube as a long coiled-coil, and its C terminus is located in the tail tip complex[20], at the tip of which is found T5 receptor-binding protein[21]. The mere interaction of T5 with its host receptor, the outer membrane iron-ferrichrome transporter FhuA, triggers in vitro the release of DNA from the capsid[22]. Moreover, functional tails can be isolated from an amber mutant in the major capsid protein[21]. This makes T5 an attractive and well-suited system to investigate *Siphoviridae* tail structure and reorganisation induced upon DNA release. Here, we determined the structure of pb6, and show that it results in the duplication/fusion of the hexamerisation domain common to all other tubes. We also determined the structure of T5 tail tube by cryo-EM to 6 Å resolution. The fit of pb6 crystal structure in the EM density map allowed proposing a pseudo-atomic model of T5 tail tube. Comparison of the structures of T5 tail tube before and after

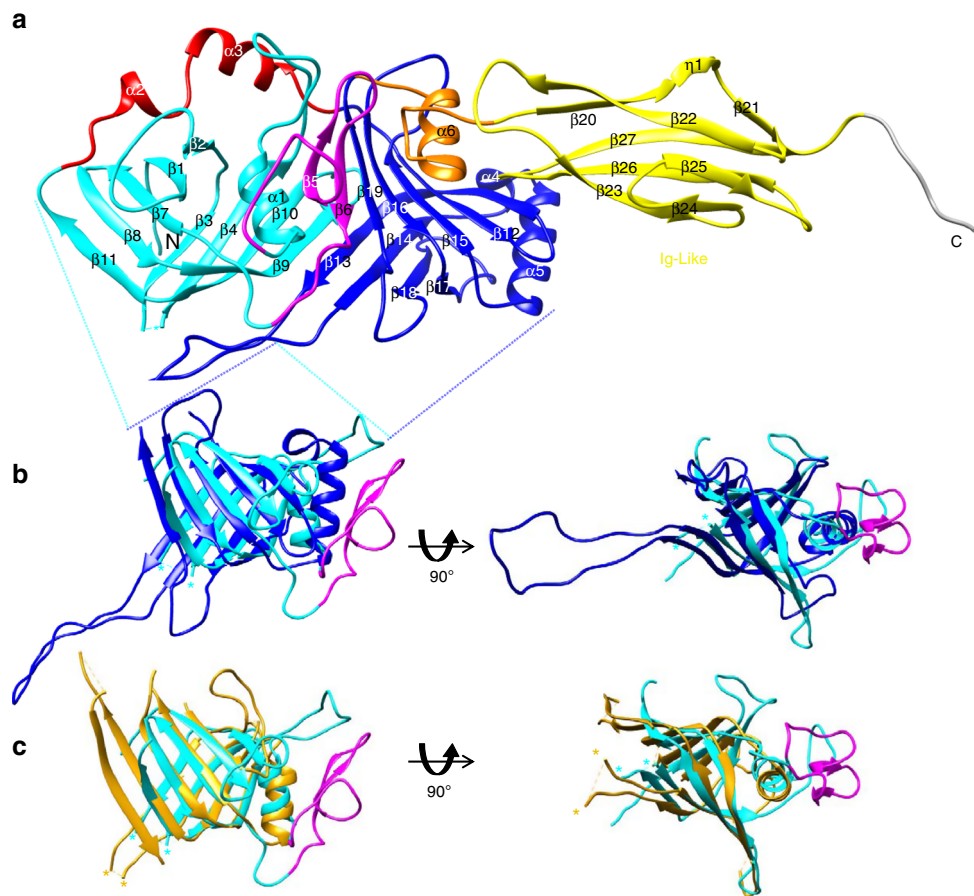

**Fig. 1** Crystal structure of pb6. **a** Ribbon representation. Cyan: subdomain 1, blue: subdomain 2, magenta: insertions with respect to all other homologous proteins in subdomain 1 (see Supplementary Fig. 1b), red: linker between the two subdomains, yellow: Ig-like domain, orange: linker between subdomain 2 and the Ig-like domain, grey: TEV cleavage site. N and C: N and C terminus, respectively. **b** Superimposition of the two pb6 subdomains. **c** Superimposition of subdomain 1 (cyan) with the inner tube protein of the T6SS of *Yersinia pestis* (gold, PBD 3V4H). Asterisks point to the departure of the long disordered 28 amino acid loop β3–β4 of pb6 and of the tube protein of *Y. pestis* T6SS

**Table 1 Crystallography data collection and refinement statistics**

| | Native PDB 5NGJ | MAD data set | | |
|---|---|---|---|---|
| | | | | |
| *Data collection* | ESRF/BM30A | | | |
| Space group | $P12_11$ | | | |
| Cell dimensions | | | | |
| *a, b, c* (Å) | 76.79, 115.08, 83.36 | | | |
| α, β, γ (°) | 90.00, 111.88, 90.00 | | | |
| | | Peak | Inflection | Remote |
| Wavelength (Å) | 0.97965 | 0.97932 | 0.97952 | 0.97836 |
| Resolution (Å) (last resolution shell) | 57.54–2.20 (2.32–2.20) | 77.80–3.00 (3.16–3.00) | 77.83–3.00 (3.16–3.00) | 57.93–3.00 (3.16–3.00) |
| $R_{merge,}$ | 0.070 (1.724) | 0.104 (0.507) | 0.111 (0.697) | 0.115 (0.683) |
| $I/\sigma(I)$ | 14.4 (1.6) | 14.6 (3.5) | 13 (2.6) | 5.4 (1.5) |
| Completeness (%) | 99.7 (97.9) | 99.7 (98.3) | 99.8 (98.4) | 99.8 (98.5) |
| Multiplicity | 7.5 (7.5) | 7.5 (7.5) | 7.6 (7.5) | 3.8 (3.7) |
| *Refinement* | Buster-TNT | | | |
| No. reflections | 67,830 (6631) | | | |
| $R_{work}/R_{free}$ | 0.2050 (0.2451)/0.2350 (0.2756) | | | |
| No. atoms | 6917 | | | |
| Protein | 6684 | | | |
| Ligand/ion | 5 | | | |
| Water | 228 | | | |
| *B* factors | 85.64 | | | |
| Protein | 86.01 | | | |
| Ligand/ion | 79.61 | | | |
| Water | 75.19 | | | |
| r.m.s. deviations | | | | |
| Bond lengths (Å) | 0.014 | | | |
| Bond angles (°) | 1.74 | | | |
| Ramachandran favoured (%) | 95.93 | | | |
| Ramachandran allowed (%) | 3.72 | | | |
| Ramachandran outliers (%) | 0.35 | | | |

interaction with its receptor shows no differences, suggesting that pb6 plays no role in the transduction of receptor binding from the tip of the tail to the capsid.

## Results

**Crystal structure of pb6**. We have determined the pb6 monomer structure at 2.2 Å resolution (Fig. 1a and Table 1). At first sight, the structure can be divided into two domains encompassing 374 and 85 residues (Fig. 1a). The C-terminal domain (residues 375–462) possesses an immunoglobulin-like (Ig-like) fold of the Big-2 family[23], confirming a previous sequence analysis[21]. Ig-like domains are very common in phage proteins and have been proposed to play accessory roles in the infection process, probably by binding to carbohydrates[24]. They are particularly found in TTPs of siphophages, as in phage λ[23, 25] and SPP1[26], where it was shown that they are dispensable for phage assembly and infectivity. In λ, however, its absence has an influence on burst size and temperature sensitivity of the phage particle[23]. For pb6, formation of tubes occurs even when this domain is absent (see below), and some T5-like TTP lack it (Supplementary Fig. 1a). A closer examination of the N-terminal domain structure reveals subdomain duplication, which is confirmed by DALI pairwise comparison[27] (Fig. 1a, b and Supplementary Table 1). The common core shared by both subdomains consists of a β-sandwich flanked by an α-helix, and a long loop. This loop is not resolved in subdomain 1 (loop β3–β4), whereas it is stabilised by crystal contacts in subdomain 2 (loop β13–β14; Fig. 1b and Supplementary Fig. 2). On a sequence level, T5-like TTPs have no homologues in the databases according to the PSIBlast and HHPRED software tools. However, both subdomains display high structural homology with TTPs of other sipho- and myophages, distal tail proteins of siphophages, T6SS tube proteins, the tube

protein of R-pyocin, baseplate hub proteins from myophages and spike proteins from T6SS (Fig. 1c, Supplementary Fig. 1b and Supplementary Table 1). Thus, T5 TTP is not an outlier in the family of TTPs, as it contains the same hexamerisation domain as other phages, T6SS and pyocin tube proteins, but results from a duplication/fusion: the unusual threefold symmetry of the tube is in fact pseudo-hexameric. Such a duplication has been observed before in myophage hub and T6SS spike proteins[14, 28]. However, these should be considered as two independent events, since the two subdomains are linked differently in pb6 and in the hub and spike proteins (Supplementary Fig. 3).

**Pseudo-atomic structure of T5 tail tube**. In parallel, we have determined the structure of T5 tail tube by cryo-EM at 6.2 Å resolution (Fig. 2a, d, g, j, Supplementary Fig. 4a and Supplementary Table 2). This structure shows a 90-Å large tube (pitch = 40.6 Å, twist = 39.1°) that is decorated by a bulb of density pointing outwards, as is often observed for *Siphoviridae* (ref. [29] and references therein). Electron density is present in the 40-Å wide lumen of the tube, which most likely corresponds to the TMP pb2. However, no structural information can be derived from this density as threefold symmetry and averaging along the tube will have blurred it. The X-ray structure of the pb6 monomer fits remarkably well in the tube density (Chimera[30] correlation 0.89 using a pb6 map simulated from atoms at a resolution of 6 Å, 879/6657 atoms being outside the contour), except for the N terminus, the long β13–β14 loop, loop β17–β18 and helix α5, all positioned at interfaces between monomers (Fig. 3a–c). Interestingly, these elements have a high temperature factor—i.e., are more flexible—in the crystal structure, and we could model them and loop β3–β4 in the unassigned EM densities observed in their vicinity, leading to a complete pseudo-atomic model of the T5 tail

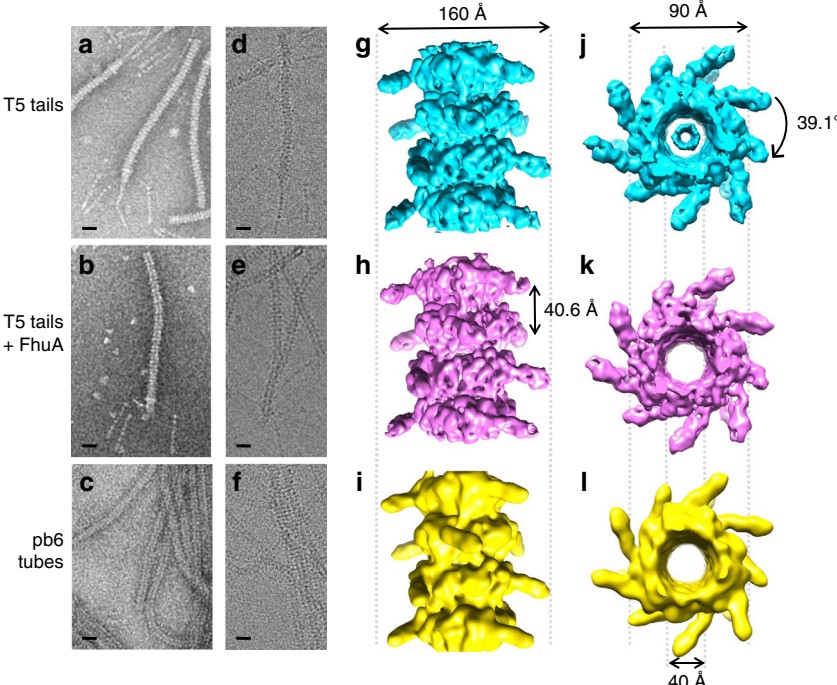

**Fig. 2** EM reconstructions of T5 tail before and after interaction with FhuA. Negative stain (**a**, **b**, **c**) and cryo-EM (**d**, **e**, **f**) images, and cryo-EM 3D reconstruction (side views (**g**, **h**, **i**) and top view (**j**, **k**, **l**)) of T5 tails before (6.2 Å resolution, **a**, **d**, **g**, **j**) and after (5.8 Å resolution, **b**, **e**, **h**, **k**) interaction with FhuA, and of pb6 tubes (8.8 Å resolution, **c**, **f**, **i**, **l**). Scale bar, 20 nm

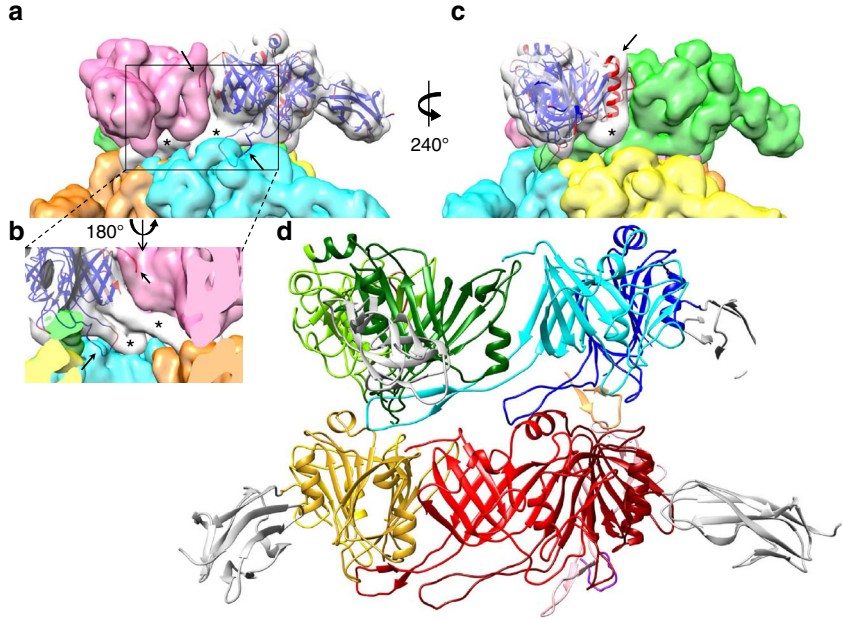

**Fig. 3** Rigid fit of pb6 crystal structure in the EM reconstruction. **a** Side view of the segmented EM tail tube density. In the white segment, the crystal structure of the pb6 monomer was rigidly fitted. It is represented as a ribbon coloured according to the averaged residue temperature factor (red: more flexible, blue: more rigid). **b** Enlarged-up view, seen from the inside of the tube and **c** view rotated 240° with respect to **a**. Unattributed densities are indicated by an asterisk, whereas arrows point to structure elements that do not fit the segmented density. **d** Side view of the pseudo-atomic model of the tail tube after flexible fitting of pb6 into each segment (see also Supplementary Fig. 5). Within each pb6 protein, the two subdomains are coloured with two tones of the same colour (light and dark green, cyan and blue, and sandy brown for the upper ring and yellow and gold, light and dark red, and salmon and purple for the lower ring), whereas the Ig-like domain is coloured white in all cases

tube (Figs. 3d and 4a and Supplementary Fig. 5). Mutation of conserved charged amino acids within these loops (Supplementary Fig. 1a), as well as the deletion of the two long loops (β3–β4 and β13–β14) or the N terminus, impaired tube formation (Fig. 5, Supplementary Table 3 and see below), confirming their importance in tube formation/stabilisation. This is a common phenomenon among phage proteins, where oligomerisation of a protein is regulated by interaction with its partner, and unstructured regions play a central role in regulating assembly[10]. In the tube assembly, a monomer is connected to six

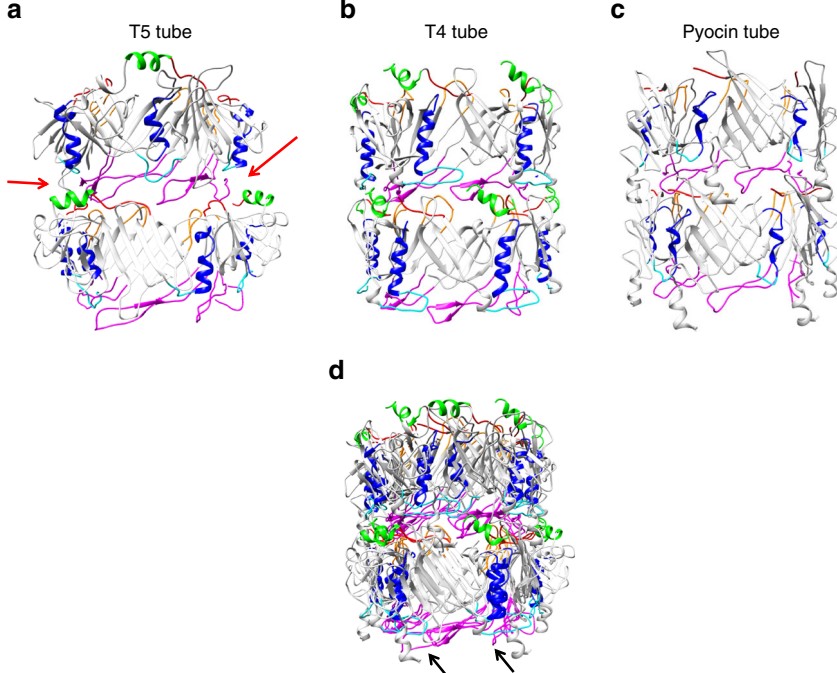

Fig. 4 Comparison of known tube structures. Side views of a stack of two TTP rings are shown, as a ribbon representation. **a** T5 tail tube pseudo-atomic model. pb6 Ig-like domain and its linker were removed for clarity. **b** Structure of the T4 tail tube (PDB 5IV5). **c** Structure of R-pyocin tube (PDB 3J9Q). Red arrows point to notches in the T5 tube that could explain tube flexibility; in T4 and pyocin tubes, additional inter-ring contacts are observed at these positions. **d** Superimposition of the three tubes, aligned on the lower ring. The longer loops of the two pb6 subdomains (black arrows) and asymmetry of the interface of two subdomains seem to impose a larger twist: T5 ring is not in register with the two other tubes in the upper ring. In **a**–**d**, similar structural features are coloured accordingly: the long loop (magenta) and helix (blue) of the common core fold, the N terminus (red) and two smaller loops (cyan and orange) and an additional N-terminal helix (green)

neighbouring subunits, making this structure very stable (pb6 tube melting temperature = 83 °C vs. 50 °C for pb6 monomer; note that it is 90 °C for purified tails). A comparison with the other tube structures available (phage T4 (PDB 5iv5[4]) and R-pyocin (PDB 3j9q[15])) indicates that the inner and outer tube diameters are identical, as are the distances between the rings. As expected from the common fold of the ring building block in these three tubes, common features in monomer interfaces can be identified (Fig. 4). However, the twists of the tubes differ (39.1° vs. 18.3° × 2 = 36.6° for pyocin and 17.9° × 2 = 36.8° for T4, Fig. 4d), which could stem from the long loops being longer in pb6. Flexibility of the T5 tail tube could stem from fewer inter-ring interactions (red arrows, Fig. 4a). The interior of the tube is strongly negatively charged, as expected for a tube that channels DNA[31].

**No structural change upon interaction with the receptor.** We then determined the structure of the tail tube after receptor binding, which results in an empty tail tube[21], by cryo-EM to 5.8 Å resolution (Fig. 2b, e, h, k, Supplementary Fig. 4a and Supplementary Table 2). We did not observe any structural differences at this resolution with the tail before receptor binding (Chimera correlation 0.98) besides the absence of the internal electron density assigned to the TMP (Supplementary Fig. 6a and Supplementary Movie 1). The resolution of both structures reached a limit around 6 Å; addition of more particles in the refinement was ineffective in improving it (Supplementary Fig. 4a). This is probably linked to the flexibility of both the Ig domain and the tail, which would limit the overall resolution. Local resolution is estimated between 5 and 6 Å for the core of the structure (subdomains 1 and 2), while the more peripheral densities (Ig-like

domains) are less well defined (resolution >6.5 Å) (Supplementary Fig. 4b).

When overexpressed, pb6 is recovered in part in the soluble fraction of broken cells, but also, in majority, in the insoluble fraction. Whereas the soluble protein is monomeric, the protein in the insoluble fraction forms very long tubes of several μm (Fig. 5a). The cryo-EM structure of these tubes at 8.8 Å resolution (Fig. 2c, f, i, l, Supplementary Fig. 4a and Supplementary Table 2) does not show any difference with both the full or empty tail tubes (chimera correlation 0.99, Supplementary Fig. 6b and Supplementary Movie 2). Thus, even though pb6 monomers can be concentrated at least up to 100 mg mL⁻¹ without forming tubes or aggregating, they can spontaneously assemble during cell lysis to form a tube that is structurally identical to that of the tail, although much longer in the absence of TMP. However, the trigger for this polymerisation is unclear, although we noticed that it is enhanced by low pH and high salt concentration. Several pb6 deletions and point mutants were overexpressed to evaluate their ability to form tubes (Fig. 5b, c and Supplementary Table 3). The design of these mutants was based on sequence conservation within T5-like TTPs (Supplementary Fig. 1a) or on monomer interfaces identified in the pseudo-atomic tail structure. As expected, and as observed in tube proteins of phage λ[13], phage SSP1[32] and *Burkholderia pseudomallei* T6SS[33], deletions of/ substitutions in the long loops abolish tube formation, confirming their importance in tube stabilisation. Tube formation is also totally or partially impaired in the case of alanine substitutions at monomer–monomer interfaces. In the latter case, tubes are still observed (Supplementary Table 3), indicating that the large interaction surfaces stabilising the tube allow some tolerance for localised perturbation.

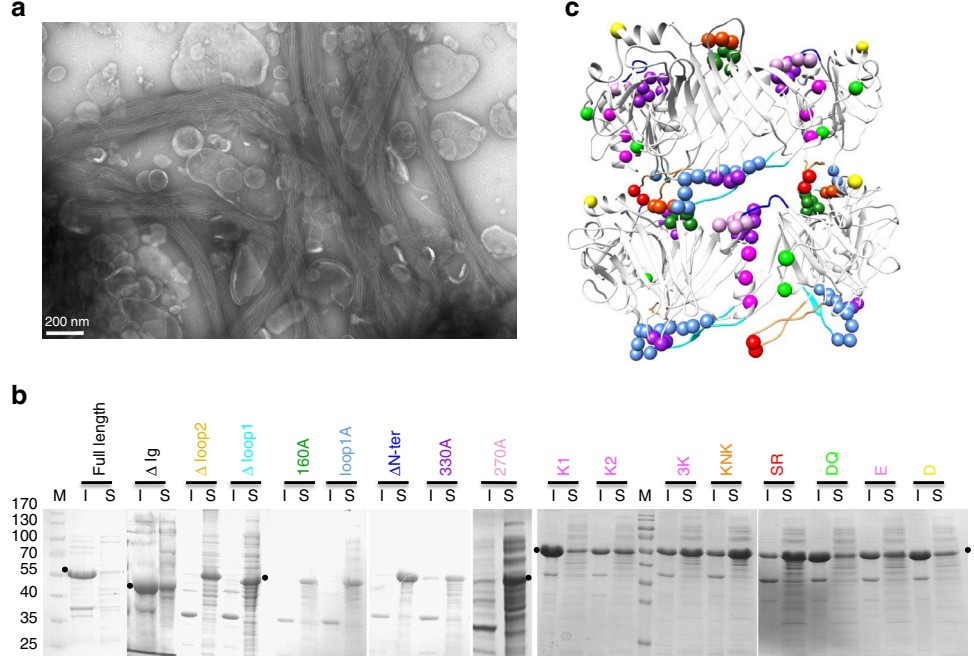

**Fig. 5** Mutations affecting tube formation. **a** Negative stain EM of the pellet of broken cells showing micrometer-long pb6 tubes. Bar, 200 nm. **b** Coomassie-stained SDS-PAGE of the soluble (S) and insoluble (I) material after centrifugation of broken cells overexpressing the full-length protein or different mutants (see also Supplementary Table 3). The black dots point to the position of pb6 in the gel. All the generated mutants are presented. M molecular markers (170, 130, 100, 70, 55, 40, 35 and 25 kDa). **c** Two rings of the T5 tail tube ribbon model, with spheres indicating the position of amino acids that were mutated. The Ig-like domain and its linker were removed for clarity. Mutations in the same colour were performed in the same mutant protein. Same colour code in **b** and **c**, see also Supplementary Table 3

## Discussion

At the resolution of our EM structures (6 Å), T5 tail tube does not undergo conformational changes upon binding of the tail tip to its receptor. Furthermore, the structure of the building block that forms the different tubes is remarkably conserved in those of sipho- and myophage tails, T6SS and R-pyocin, suggesting little or no flexibility of the structured core of the protein. We suggest that in all these structures, the TTP would have only a scaffolding function and would not be involved in signal transduction or in expulsion of the TMP. It is possible that previous contradictory observations on siphophage SPP1[11] resulted from negative-stain artefacts that may occur during stain interaction with the protein, interaction of the protein with the carbon layer and/or drying of the sample.

Hence, in *Siphoviridae*, we propose both signal transduction and membrane perforation functions are fulfilled by the TMP with the TMP constrained in a metastable state within the tail assembly, as for the sheath of contractile devices[3, 15] and for the *Podoviridae* core proteins[34]. Interaction of the receptor binding proteins with the host receptor would initiate conformational changes, culminating in the release of TMP, leading to its expulsion from the tail tube and its insertion into the cell wall[20]. Moreover, the membrane insertion property and muralytic activity described for the C-terminal domains of T5 TMP would serve as complementary forces driving TMP ejection/refolding[20]. The exit of the TMP at the level of the head-to-tail connector would in turn induce conformational changes in the latter, resulting in its opening for DNA release[35].

In *Myoviridae* and the other bacterial contractile devices, contraction of the sheath provides the energy to perforate the cell wall[3, 15, 36]. It does not, however, seem to be the signal that opens the capsid in *Myoviridae*, as contracted tails with full capsids have been isolated[3]. The signal for capsid opening was proposed to be generated by TMP ejection[2] or conformational changes in the

TTP[37]. Given the remarkable conservation of structures and tail organisation (tube, inner baseplate)[9] within *Siphoviridae* and *Myoviridae* phages, we propose that the TMP of *Siphoviridae* and *Myoviridae* plays a similar role in signal transduction.

## Methods

**Protein and tail production and purification.** Production and purification of the monomeric soluble pb6 protein bearing a hexahistidine tag at its C terminus were performed as described[38]: *E. coli* BL21 (DE3) strain (BL21(DE3) Singles™ competent cells, Merck), transformed with the *pb6*-pLIM13 plasmid was grown in LB medium with kanamycin (50 µg mL$^{-1}$) at 37 °C until stationary phase. We relied on the leak of the promoter for protein expression as induction yielded 100% insoluble protein. Cells were broken in 50 mM Tris-HCl, pH 8—which optimises pb6 monomer fraction—with a microfluidizer, and the clarified cell lysate, supplemented with 15 mM imidazole, was loaded onto a Ni-NTA column (HiTrap™ Chelating HP 5 mL GE Healthcare) equilibrated with 20 mM Tris pH 8, 100 mM NaCl, 15 mM Imidazole. The column was washed with the equilibration buffer, and proteins were eluted with 20 mM Tris pH 8, 300 mM imidazole. Protein containing fractions were pooled and diluted five times before loading onto an ion exchange column (HiTrap™ Q HP 1 mL, GE Healthcare) equilibrated with 20 mM Tris pH 8. The protein was eluted by a 0–1 M NaCl linear gradient. Seleno-metionine protein was produced in the same strain but grown in an in-house M9 minimal medium (50 mM Na$_2$HPO$_4$, 20 mM K$_2$HPO$_4$, 10 mM NaCl, 20 mM NH$_4$Cl, 2 mM MgSO$_4$, 100 µM CaCl$_2$, 2 g L$^{-1}$ glucose). Fifteen minutes prior to 100 mM IPTG induction, 30 mg L$^{-1}$ selenomethionine, 50 mg L$^{-1}$ threonine, lysine, phenylalanine and leucine, 25 mg L$^{-1}$ isoleucine and valine were added to the culture. Protein was purified as the native protein.

FhuA was produced and purified as described[39]: *E. coli* AW740 {FhuA31 ΔompF *zcb*::Tn*10* ΔompC}transformed with the pHX405 plasmid, in which the *fhuA* gene is under control of its natural promoter, was grown at 37 °C in LB medium supplemented with ampicilin (125 µg mL$^{-1}$), tetracyclin (10 µg mL$^{-1}$) and 2,2′ bipyridyl (100 µm), an iron chelator used to induce the production of the protein. After clarification of the cell lysate, total membranes were recovered by ultracentrifugation and solubilised by 50 mM Tris pH 8, 2% (w/w) OPOE (*N*-octylpolyoxyethylene, Bachem) at 37 °C for half an hour. The insoluble material was recovered by ultracentrifugation and solubilised 1 h at 37 °C by 50 mM Tris pH 8, 1 mM EDTA, 1% (w/w) LDAO (*N*,*N*-dimethyl dodecylamine-N-oxide, Anatrace). The solubilised fraction, recovered after ultracentrifugation, was supplemented with 4 mM MgCl$_2$ and 5 mM imidazole and loaded on a nickel affinity column (HiTrap™ Chelating HP 5 mL GE Healthcare) previously

equilibrated with 0.1 % LDAO, 20 mM Tris pH 8, 200 mM NaCl and washed with the same buffer. Prior to elution, a delipidation step was performed with 10 mL of 1% LDAO in the same buffer. The protein was eluted from the column with 0.1% LDAO, 20 mM Tris pH 8, 200 mM imidazole, and loaded onto an ion exchange column (HiTrap[TM] Q HP 1 mL, GE Healthcare) equilibrated with 0.05% LDAO, 20 mM Tris pH 8. The protein was eluted by a 0–1 M NaCl linear gradient in the same buffer. Purified FhuA was transferred into amphipol (A8–35) at a protein: amphipol ratio of 1:5 (w:w). After 1 h 40 min incubation of the protein:amphipol mixture at room temperature, detergent was removed by the addition of 200 mg mL$^{-1}$ BioBeads (BioRad) and incubation on a stirring wheel at room temperature for 45 min.

Tubes of pb6 tubes were purified from the insoluble fraction of the cell lysate: the broken cell pellet was resuspended and incubated for 1 h at 37 °C in 50 mM Tris-HCl pH 8, 100 mM NaCl, 0.5% Triton X-100, 1 mM EDTA, 100 μg mL$^{-1}$ lysozyme before loading onto a 30–50% sucrose gradient (20 mM Tris-HCl pH 8, 100 mM NaCl, 0.05% Triton X-100) centrifuged at 10,000 rpm for 30 min (SW41 rotor) at 4 °C. Pure tubes were recovered in the lower part of the gradient and extensively washed by dilution/centrifugation in 20 mM Tris-HCl pH 8, 100 mM NaCl to remove sucrose and detergent.

Production and purification of T5 tails were developed elsewhere[21]: E. coli F cultures at 37 °C were infected during the exponential growth phase with the amber mutant phage T5D20am30d, at a multiplicity of infection of 6–7. After complete cell lysis (OD$_{600nm}$ < 0.15), T5 tails were precipitated from the culture medium by incubation with 0.5 M NaCl and 10% (w/w) PEG 6000 overnight at 4 °C. The pellet was resolubilised in 10 mM Tris pH 8, 100 mM NaCl 1 mM CaCl$_2$, 1 mM MgCl$_2$ and purified on a glycerol step gradient (10–40 %) in the same buffer, and centrifuged 2 h at 20,000 rpm (SW41 rotor). The gradient fractions containing the tails (usually ~10 % glycerol), diluted ten times in 10 mM Tris pH 8, were loaded onto an ion exchange column (HiTrap[TM] Q HP 1 mL, GE Healthcare) equilibrated with the gradient buffer without NaCl and washed in that same buffer. The tails were eluted by a 0–0.5 M NaCl linear gradient in the same buffer.

**Mutations.** Expression vectors of pb6 mutants were obtained using the Quick-Change Lightning Site-directed Mutagenesis Kit (Agilent Technologies) on the pb6-pLIM13 plasmid[38] according to the recommendations of the manufacturer, with primers described in Supplementary Table 4. All pb6 mutants were sequenced prior to transformation. Mutant proteins were produce as wild-type protein, except for pb6-ΔNter, which production had to be induced with 1 mM IPTG. Cells were resuspended and broken in 50 mM Tris-HCl, pH 8, 150 mM NaCl, which enhances the relative amount of pb6 tubes. After ultracentrifugation of the cell lysate, soluble and insoluble fractions were deposited on SDS-PAGE to compare relative quantities of pb6.

**Measure of melting temperature.** Melting temperatures of purified pb6 monomer (3 mg mL$^{-1}$), purified pb6 tube and purified T5 tail (~6 mg mL$^{-1}$) were obtained using a Prometheus NT.48 (NanoTemper Technologies) according to the manufacturer's recommendations.

**Crystallisation.** Initial hits were obtained on the high throughput crystallisation platform on site. Crystals were reproducibly obtained at 4 °C by the hanging drop vapour diffusion setup by mixing 1:1 (v/v) protein solution (8 mg mL$^{-1}$) with reservoir buffer containing 10% 3350 PEG, 0.2 M HEPES pH 7.5, 0.3 M LiSO$_4$, 0.1 mM MnCl$_2$. Selenomethionine-pb6 crystals were obtained with a protein solution at 7 mg mL$^{-1}$ and a reservoir buffer containing 11% 3350 PEG, 0.2 M HEPES pH 7.5, 0.1 M LiSO$_4$, 40 mM MnCl$_2$. Crystals typically grew within 2 days. All crystals were cryoprotected with 25% PEG 400 and flash-frozen in liquid nitrogen using nylon loops (CryoLoop[TM], Hampton Research).

**Data collection and crystallography.** X-ray data sets were collected at the European Synchrotron Radiation Facility (French beamline for Investigation of Proteins – BM30A). A native data set was collected on a native crystal up to 2.2 Å and a MAD data set was collected on a single selenomethionine-pb6 crystal up to 3 Å. Diffraction frames were integrated using the programme XDS[40] and the integrated intensities were scaled and merged using the CCP4 programmes[41] SCALA and TRUNCATE, respectively. A summary of the processing statistics is given in Table 1. pb6 was solved de novo by combining the MAD and the native data sets. Selenium positions were determined within the asymmetric unit using the programme SHELXD[42]. Heavy-atom refinement and initial phasing were performed using the programme SHARP[43]. Phases from SHARP were improved by density modification using the programme SOLOMON[44]. The figure of merit after density modification was 0.82. The first steps of model building were performed automatically using BUC-CANEER[45] and completed manually with COOT[46]. Model refinement was performed with PHENIX[47] and BUSTER (2016 BUSTER version 2.10.3. Cambridge, UK: Global Phasing Ltd). The model was optimised through iterative rounds of refinement and model building. Automated non-crystallographic symmetry and translation/libration/screw were applied, and at the end stages of the refinement hydrogen were added (except for solvent molecules).

Refinement statistics are summarised in Table 1, and a stereo image of a portion of the electron density map is presented in Supplementary Fig. 7. The atomic coordinates and measured structure factor amplitudes for pb6 were deposited in the Protein Data Bank with accession code 5NGJ.

**Negative stain electron microscopy.** Negative-staining EM allowed purity and concentration assessment, as well as adapting the T5 tail:FhuA ratio prior to cryo-grid preparation. Negative-stain grids of all samples were prepared using the mica-carbon flotation technique over a 2% sodium silicotungstate solution and were observed with an FEI/Thermo Fisher Scientific Tecnai 12 microscope (LaB$_6$ electron source, 120 kV) equipped with a Gatan Orius 1000 CCD camera.

**Cryo-electron microscopy.** pb6 tubes and purified T5 tails incubated with or without FhuA were prepared following the same procedure. An aliquot of 3 μL of sample was applied to glow discharged 2:1 Quantifoil holey carbon grid (Quantifoil Micro Tools GmbH, Germany) and the grids were plunge-frozen in liquid ethane with a Vitrobot Mark II (FEI/Thermo Fisher Scientific). Samples were observed with an FEI/Thermo Fisher Scientific Polara at 300 kV. Images were recorded on a K2 Summit direct detector (Gatan Inc., USA) in super resolution counting mode. Movies were recorded for a total exposure of 4 s and 100 ms per frame, resulting in 40-frame movies with a total dose of ~40 e$^-$ Å$^{-2}$. For tails incubated with FhuA and pb6 fibres, the magnification was ×23,000 (0.82 Å per pixel at the sample level) while for T5 tails, the nominal magnification was ×15,500 (1.24 Å per pixel at the sample level).

**3D reconstruction.** An initial model was created by back-projecting a portion of a single straight T5 tail (total of ~6 consecutive trimeric stacks) and by applying threefold symmetry using SPIDER[48]. This initial model was refined using a classical projection matching method as described[49]. Briefly, the contrast transfer function (CTF) corrected tail images were boxed into 128 × 128 pixel windows with an overlap of ¾ of the box. The generated stacked images (1000 images) were aligned against the projections of the model and averaged into classes. The classes were then back projected to obtain a new threefold symmetrised model. After 10 cycles of refinement, we obtained a map at ~10 Å resolution in which the crystal structure of pb6 could be fitted without ambiguity. In order to further improve the resolution, a larger data set was processed as described below.

Images were first motion corrected either with Digital Micrograph (Gatan Inc., USA) for T5 tails or with MotionCorr[50] for tails incubated with FhuA and pb6 tubes. The latter were also binned two times by Fourier cropping before further processing (final pixel size of 1.64 Å per pixel). The same image analysis procedure was followed for all samples. Tubes were first boxed as overlapping segments with boxer[51]. The interbox distance was set to one axial rise (z = 40 Å) for all samples. The coordinates were then imported into RELION 1.4 [52]. CTFFIND4[53] was used to determine the CTF parameters. 2D classification was then performed in RELION to remove outliers. The low-resolution 3D initial model (see above) or a featureless cylinder were used as the initial reference for a first 3D refinement imposing C3 symmetry. As nearly all possible helical segments were picked by choosing an interbox distance equal to the axial rise, no further helical averaging was needed. This was confirmed later by two different refinement strategies, both enforcing helical symmetry, using IHRSR[54] and RELION 2[55]. Both approaches gave 3D density maps nearly identical to the one obtained without helical symmetry empty and full tails were then separated by successive 3D classifications. 3D classes displaying central density were grouped together while those having an empty channel formed a second group. A last 3D refinement was then done on the 22704 empty (T5 tail incubated with FhuA) and the 24243 full (T5 tails) particles, which yielded 3D reconstructions at 5.8 and 6.2 Å, respectively (Fig. 2g, j, h, k and Supplementary Table 2) after post processing in RELION with a threshold mask. For the pb6 tubes, no 3D classification was needed as the tubes were all empty, and the final 3D reconstruction had 8.8 Å resolution and included 12,244 particles (Fig. 2i, l and Supplementary Table 2). Map visualisation, segmentation and fitting were done with CHIMERA[30].

As the 3D reconstructions obtained for full or empty T5 tails are virtually identical at the pb6 level, the two data sets were combined by scaling the T5 tails acquired at the nominal magnification of 15,500 from a pixel size of 1.24–1.64 Å per pixel. The combined data set was processed as described above but with RELION 2, which implements helical symmetry during image processing. For consistency reasons, the data sets for full and empty T5 tails were re-processed with RELION 2. As expected, the 3D reconstructions for both samples were nearly identical to the one obtained with RELION 1.4. The final 3D reconstruction for the combined data set included 55,447 particles and reached 6 Å resolution at FSC = 0.143, but with enhanced interpretability, notably at the subunit interfaces (Supplementary Fig. 4). This map was used to develop a pseudo-atomic model of the pb6 tube.

**Model building.** The crystal structure of pb6 was rigidly fitted in the tube density using the 'fit in map' option in CHIMERA[30]. This original fit was used to guide segmentation of the map using SEGGER[56]. Isolated densities corresponding to a monomer was then used to build the missing loop (loop β3–β4, residues 38–67) using COOT[46] and model the structural features rearranged in the tube assembly using FLEXEM[57]. Definition of rigid domains was achieved using the RIBFIND server[58], and the output file was slightly edited to allow mobility in protein regions not

correctly fitting density in the original rigid fit. For the most mobile element (N terminus 1–8, the two long loops β3–β4 and β13–β14 and loop β17–β18) different conformations were used as starting points (manually modelled in COOT) and subjected to 5–10 iterations of simulated annealing rigid body (FLEXEM Molecular Dynamics (MD) mode). A new set of starting points was built by combining the best-looking models of all mobile elements (typically twisted or straight for the loops). These new starting points were subjected to five iterations of MD refinement. The convergence of the models generated from different starting points lowered the uncertainty of the orientation of the polypeptide chain.

**Data availability**. Coordinates and structure factors for pb6 have been deposited in the Protein Data Bank under accession number 5NGJ. Electron microscopy maps have been deposited in the Electron Microscopy Data Bank under accession codes EMD-3689 (full tail tube), EMD-3689 (empty tail tube), EMD-3691 (pb6 tube) and EMD-3692 (combined tail tube). Other data supporting the findings of this study are available from the corresponding author on reasonable request.

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

## Acknowledgements

We thank Flavien Grégoire, Alice Labaronne and Valentin Sencio for help at different stages of the project, Daphna Fenel for help with negative-stain EM, Christine Ebel for support and James Conway for proof reading of the manuscript. We also acknowledge access to the European Synchrotron Radiation Facility (ESRF) and SOLEIL beamlines for crystal screening, and thank scientists of the FIP-BM30A at ESFR, in particular Michel Pirrochi for help with data collection. This work used the electron microscopy and the high-throughput crystallography platforms from the Grenoble Instruct Center (ISBG: UMS 3518 CNRS-CEA-UJF-EMBL) with support from FRISBI (ANR-10-INSB-05-02) and GRAL (ANR-10-LABX-49-01) within the Grenoble Partnership for Structural Biology (PSB). The IBS Electron Microscope facility is supported by the Rhône-Alpes Region, the Fonds Feder, the Fondation pour la Recherche Médicale and GIS-IBiSA. We acknowledge funding by the ANR PerfoBac (ANR-16-CE11-0027-01). CAA was funded by GRAL.

## Author contributions

C.-A.A., E.G., G.S., and C.B. designed and conceived the experiments. C.-A.A., G.E., S.E., C.V., M.B., P.B., E.G., G.S., and C.B. performed experiments (C.-A.A.: biochemistry, molecular biology, crystallogenesis and crystallography, E.M. image processing, C.V.: molecular biology, S.E. and E.G.: crystallography, M.B. and G.S.: E.M. data acquisition, G.E., G.S. and C.B.: E.M. image processing, P.B.: designed the purification of T5 tails). C.-A.A., G.E., S.E., E.G., G.S. and C.B. analysed the data. C.B. wrote the paper with contributions from C.-A.A., G.E., E.G. All authors revised and approved the manuscript.

## Additional information

**Competing interests:** The authors declare no competing financial interests.

