## [Peer Review File · Nature Communications]

Reviewers' comments:

Reviewer #1 (Remarks to the Author):

The manuscript by Arnaud et al. reports investigations on a siphophage (phage T5) tail structure, before and after DNA release. From that, they speculate on the mechanism of DNA release in Siphoviridae. Whereas a wealth of reports has targeted myophages, such as phages T4 ormu, leading recently to the complete atomic structure of phage T4 baseplate and tail, our knowledge of siphophage lacks behind. The only structure of a siphophage tail protein (TP), that of phage lambda has been determined by NMR, and 1/3 of the structure is lacking. Furthermore, besides a few low resolution tail structures of lactophages and phage SPP1, determined by negative staining electron microscopy (nsEM), nothing was known at atomic resolution.

Here, the authors determined the crystal structure of T5 TTP at 2.2 Å resolution, and identified 3 domains (already suspected by in-silico approaches) : 2 domains belong to the TTP fold observed in phage Lambda, T4, pyocins, and T6SS Hcp, and form the core of the tube. They are very similar and therefore the 3 double-TP tube ring of T5 is totally comparable of the hexameric assembly of the other – most common –siphophages. The 3d domain possess an Ig fold, often observed in siphophage and with a known structure from phage lambda. However, this is the first report of a complete structure of a siphophage TTP.

Then, they determined the cryoEM structure of T5 tail at near-atomic resolution (~6Å) for the tube before DNA ejection, i.e. containing the tape measure protein (TMP), after DNA ejection, i.e. without the TMP (tube empty), and in a 3d artificial state resulting from in-vitro tail polymerization. In brief, the three tube structures are exactly the same, besides the presence of TMP in the 1st one.

The combination of the TTP crystal structure and the near-atomic resolution of the tail tube, made it possible to generate a pseudo atomic structure of the tail with full confidence, revealing the role of the loops in the TTP bricks association within the tube. Finally, all the inter-TP interactions, leading to tube assembly, have been confirmed by loop deletion and single-residue mutagenesis – 17 in total – reinforcing the pseudo-atomic model.

In conclusion, the results are sound and well presented.

However, I have two concerns : presentation could be improved, a couple of figures essentially, and the discussion and the presentation of a mechanistical model is subject to caution, in my view.

Points to be improved :

-The title is too affirmative. I will develop it later in my review. More realistic would be : Bacteriophage T5 tail tube structure suggests a trigger mechanism for Siphoviridae DNA ejection.

- Lines 20-21 : « We rather suggest that the TMP, folded in a metastable state and destabilized by host binding, would transmit the signal. »

I do not agree with this statement, and will develop later why. Besides which signal is transmitted ? from where to where ?

- Line 43 : 40 unusual trimers. It should be explained why and how they are unusual. This can be done using previous reported results.

- Lines 55-56/ Ig domains have been shown to be important in phage lambda. Depletion of these domains lead to a marked infectivity decrease.

- Lines 98-99 : the interior of the tube is negatively charged ; OK. Any clue on the charge of the TMP ? TMP being helical it would be possible to see if it can oppose a positive patch to the TTP.

- A Figure (in Fig 2 or 3) would be welcomed that illustrates clearly the contacts between the TTP subunits and loops. Something is presented in Ext. Fig. 5, but it is not very clear, and such a figure would be important in the main text.

- Figure 3. I personally can see nothing in this figure. The EM map makes the ribbon almost invisible. I would not show the ribbon/map fitting, or in a supplementary figure.

- Figure 4 c. The sticks view of the mutated residues is useless and obscures the figure. Replace each

residue representation by a colored sphere.

Now, concerning the TMP role (abstract and lines 142-145, and lines 147-149).

There was two hypothesis concerning the transmission of the signal arising from receptor binding to the connector : 1/ a TTP conformational change, moving through a domino-effect from the tail tip to the connector (Plisson et al, EMBO J) and 2/ a transmission by the TMP. I totally agree with the present findings that disprove the first mechanical hypothesis, and place the TMP as the partner of signal transmission.

However, TMP is predicted as a long helix, except for two small domains at its N- and C- termini, facing the connector and the baseplate/tail-tip, respectively. Therefore, overall, TMP has no 3D fold, metastable or not. The first trigger occurs when a siphophage bind to its receptor using either its baseplate (saccharidic) or its tail-tip (protein). This may result in a conformational change of the baseplate/tail-tip (observed in the case of some lactococcal phages) providing a mechanical pull to the TMP. Then, as proposed by the authors, the retraction of the TMP would provoke the connector opening. Therefore the TMP transmits a signal from the anti-receptor to the connector. It is not a « trigger » but a « transmitter ».

Finally, the siphoviridae mechanism for DNA release might be totally different between siphoviridae and myoviridae. In the latter case, the baseplate conformational change activates the sheath contraction and expels the tail tube. The TMP is not involved in this mechanism. To my knowledge, the trigger for TMP release and DNA ejection in Myoviridae is not documented, but may be also linked to the large mechanical stress occurring during tail contraction.

Reviewer #2 (Remarks to the Author):

Summary.

Arnaud et. al determined the crystal structure of the tail tube protein of phage T5 by X-ray crystallography, analyzed the structure of phage tails with the help of electron microscopy, and performed mutagenesis to address some aspects of tail assembly. The structures of the phage tail in its pre- and post- host cell attachment states have been determined. It showed that the protein occupying the central channel of the tail leaves this channel upon receptor binding. Recombinantly expressed tail protein pb6 assembles into a structure that is virtually identical to that of the post-attachment empty tail. Combining all the observations and taking into account similarities of non-contractile and contractile tails, the authors put forward a hypothesis that the tape measure protein is "stored" in the tail's channel in a metastable state that promotes its expulsion upon interaction of the tail with a receptor on the host cell surface.

Major concerns.

None.

Minor concerns.

The MS will benefit from reading by a colleague who (hopefully) will be able to pick up additional technical and language imperfections besides the ones given below.

Lines 27-30. These are well studied for the long, contractile tailed Myoviridae¹⁻⁴ and short tailed Podoviridae⁵⁻⁸. For the large family of long flexible tailed Siphoviridae, the mechanism by which receptor binding, occurring at the tail distal end, promotes DNA ejection remains unclear.

To the contrary, people working on Myo- and Podoviridae believe that "the mechanism, which receptor binding, occurring at the tail distal end, promotes DNA ejection" is much much clearer in Siphoviridae considering that NOT A SINGLE INSTANCE of tail tube receptor is known for Myoviridae or Gram-negative Podoviridae.

Line 50. New paragraph.

Line 63. Although T5-like TTPs have no homologues in the databases...

I do not understand. I thought you already made a pretty convincing case that pb6 is TTP. All TTPs are homologous. Please rephrase or clarify. Did you mean that you cannot see this relationship from the analysis of the amino acid sequence alone? What about HHpred?

Lines 69-70. Such a duplication has been observed before in myophage and T6SS hub proteins. The correct reference here is <http://www.nature.com/nature/journal/v415/n6871/abs/415553a.html>
This have been known for about 10 years prior to ref. 12...

Line 76-79. Electron density is present in the 40-Å wide lumen of the tube, due to the presence of the TMP pb2. However, no structural information can be derived from this density as three-fold symmetry and averaging along the tube will have blurred it. And _helical_ symmetry?

As you cannot be sure what you actually see insider the channel, you have to be more careful in interpreting this density: Electron density is present in the 40-Å wide lumen of the tube, which most likely corresponds to TMP pb2.

Line 80. Chimera correlation 0.89

This does not make sense for people not familiar with UCSF Chimera (reference needed anyway). If you give the correlation coefficient here, you have to give additional parameters - the radius around the atoms, or perhaps, how many atoms (out of all) are in the density).

Line 82. Remarkably, these elements

"Remarkably" is a wrong word. Besides, you have remarkably two lines above (line 80). Interestingly will be more appropriate here.

Lines 95-97. However, the twist of the tube differs, which could stem from the long loops being longer in pb6 (Extended Data Fig. 5d). So, what are the helical parameters and how they compare with those of the pyocin/T4 tube?

Line 99. New paragraph.

Extended Data Table 3: Summary of the performed mutants. p. indicates punctual protein mutations. You probably meant "point mutations" not "punctual", correct?

Reviewer #3 (Remarks to the Author):

In this study by C-A Arnaud and colleagues, the authors have used X-ray crystallography and TEM to study the bacteriophage T5 tail tube structure with a goal to elucidate the mechanism behind DNA ejection. The authors solved the 2.2Å resolution structure of the TTP and the empty and FhuA (host receptor)-interacted tail tube at 6-9Å resolution. This is a robust piece of work and could be published subject to attending to the comments/criticisms and questions provided below.

Main text

1. The author's argument that the empty and FhuA-including tubes are effectively identical is constrained by the low 8Å resolution reconstructions. Is that resolution sufficient to claim that there is indeed no conformational change and that mere interaction of the tube tip to host receptor instigates signal transduction?

This referee strongly feels that a higher resolution (e.g. near atomic resolution) and/or other complementary biochemical/mutational studies is needed and the authors should soften their stance in

the title, abstract and the last paragraph – a central theme of their study and the phrase “.elucidates trigger..” in title is clearly overambitious.

2. Are there other examples tail tube formed by trimers. Indeed the two pseudo similar subdomains in pb6 structure effectively leads to a traditional hexameric tube protein assembly as in many other tail tubes.

3. Could the author further comment why the loops at the N-terminus are essential for tube formation even though they find these loops to be disordered in the crystal structure?

Material and Methods

1. Line 165: After 1 h and 40 minutes...

2. Line 234 – this first sentence is unclear. What does 6 repeats stand for? How exactly was the back projection carried out? Overall why was a initial model based on negative stain analysis necessary to include as input in RELION. Did the author try a featureless hollow cylinder as a starting model to check any model bias?

3. The authors state that setting the separation of overlapping boxes to $\sim 40\text{\AA}$ (axial rise) will allow all helix views to be available. How can the authors be sure of that? Why could not they use IHRSR or SPRING to ensure helix symmetry is applied completely.

4. The figures are uniformly of poor quality and should be redone. For instance Extended Data Figure 6 is particularly poor as the different colours are barely distinguishable In the view shown in Fig. 1a Nt end is not visible. Fig.2 should be recast with panel numbers 1,2, 3,and 4 – the current assignment mentioned in the legend is confusing.

There are too many bad grammar and peculiar English usages. It is mandatory to have their manuscript properly proof read (egregious errors e.g. what is peptidic in line 282, PDB not PBD in Extended Data Table 2).

Point-to point response to reviewers

We are grateful for the reviewers' constructive comments, corrections and suggestions and appreciate the efforts made to improve the manuscript. We largely agree with the points raised, and considered all but one. A native English speaker colleague corrected the revised version of the manuscript.

In the following, reviewers' comments and questions are in plain and black text and *our response in italics and blue*. When significant changes in the text were made, "*the paragraph has been copied around quotes and the added or changed text in blue*".

Reviewer #1 (Remarks to the Author):

The manuscript by Arnaud et al. reports investigations on a siphophage (phage T5) tail structure, before and after DNA release. From that, they speculate on the mechanism of DNA release in Siphoviriae. Whereas a wealth of reports has targeted myophages, such as phages T4 or mu, leading recently to the complete atomic structure of phage T4 baseplate and tail, our knowledge of siphophage lacks behind. The only structure of a siphophage tail protein (TP), that of phage lambda has been determined by NMR, and 1/3 of the structure is lacking. Furthermore, besides a few low resolution tail structures of lactophages and phage SPP1, determined by negative staining electron microscopy (nsEM), nothing was known at atomic resolution.

Here, the authors determined the crystal structure of T5 TTP at 2.2 Å resolution, and identified 3 domains (already suspected by in-silico approaches) : 2 domains belong to the TTP fold observed in phage Lambda, T4, pyocins, and T6SS Hcp, and form the core of the tube. They are very similar and therefore the 3 double-TP tube ring of T5 is totally comparable of the hexameric assembly of the other – most common – siphophages. The 3d domain possess an Ig fold, often observed in siphophage and with a known structure from phage lambda. However, this is the first report of a complete structure of a siphophage TTP.

Then, they determined the cryoEM structure of T5 tail at near-atomic resolution (~6Å) for the tube before DNA ejection, i.e. containing the tape measure protein (TMP), after DNA ejection, i.e. without the TMP (tube empty), and in a 3d artificial state resulting from in-vitro tail polymerization. In brief, the three tube structures are exactly the same, besides the presence of TMP in the 1st one.

The combination of the TTP crystal structure and the near-atomic resolution of the tail tube, made it possible to generate a pseudo atomic structure of the tail with full confidence, revealing the role of the loops in the TTP bricks association within the tube. Finally, all the inter-TP interactions, leading to tube assembly, have been confirmed by loop deletion and single-residue mutagenesis – 17 in total – reinforcing the pseudo-atomic model.

In conclusion, the results are sound and well presented.

However, I have two concerns : presentation could be improved, a couple of figures essentially, and the discussion and the presentation of a mechanistical model is subject to caution, in my view.

Points to be improved :

-The title is too affirmative. I will develop it later in my review. More realistic would be : Bacteriophage T5 tail tube structure suggests a trigger mechanism for Siphoviridae DNA ejection.

Corrected as suggested

- Lines 20-21 : « We rather suggest that the TMP, folded in a metastable state and destabilized by host binding, would transmit the signal. »

I do not agree with this statement, and will develop later why. Besides which signal is transmitted ? from where to where ?

We agree this is indeed only a hypothesis. This part of the abstract was removed.

- Line 43 : 40 unusual trimers. It should be explained why and how they are unusual. This can be done using previous reported results.

The previous sentence in the text states that phage tail, T6SS and pyocin tubes have been shown to be formed of hexamers. We have added a sentence, based on previous results (as already referenced), stating that the tube of T5 tail is formed of trimers, which thus is unusual. We have also added a sentence concerning the only other phage that is known to have a three-fold symmetry in its tail:

“Phage T5 is a Siphoviridae infecting the Gram-negative host Escherichia coli. Its overall structure has been determined¹⁶, showing an unusual three-fold symmetry of the tail tube¹⁷. To our knowledge, such a three-fold symmetric tail tube has been observed in only one other case, for siphophage ΦCbK^{18,19}.”

- Lines 55-56/ Ig domains have been shown to be important in phage lambda. Depletion of these domains lead to a marked infectivity decrease.

Indeed. We had shortened this paragraph in the first version of the paper due to size limit. Because this is not the case any longer, we have elaborated, citing the work done on SSP1 and λ:

“Ig-like domains are very common in phage proteins and have been proposed to play accessory roles in the infection process, probably by binding to carbohydrates²¹. They are particularly found in TTPs of siphophages, as in phage λ^{23,25} and SPP1²⁶, where it was shown that they are dispensable for phage assembly and infectivity. In λ, however its absence has an influence on burst size and temperature sensitivity of the phage particle²³.”

- Lines 98-99 : the interior of the tube is negatively charged ; OK. Any clue on the charge of the TMP ? TMP being helical it would be possible to see if it can oppose a positive patch to the TTP.

We have analysed the primary sequence of the coil-coiled region of pb2. Negatively and positively charged residues are uniformly distributed along the sequence, and in the absence of a structure, it is difficult to say whether there will be positive patches interacting with the negatively charged pb6 inner tube.

- A Figure (in Fig 2 or 3) would be welcomed that illustrates clearly the contacts between the TTP subunits and loops. Something is presented in Ext. Fig. 5, but it is not very clear, and such a figure would be important in the main text.

We have made a new figure 3 were the model of the tube is shown more clearly.

- Figure 3. I personally can see nothin in this figure. The EM map makes the ribbon almost invisible. I would not show the ribbon/map fitting, or in a supplementary figure.

We have completely restructure figure 3 that is now hopefully clearer.

- Figure 4 c. The sticks view of the mutated residues is useless and obscures the figure. Replace each residue representation by a colored sphere.

A new representation has been added with a coloured sphere in place of each mutated residue.

Now, concerning the TMP role (abstract and lines 142-145, and lines 147-149). There was two hypothesis concerning the transmission of the signal arising from receptor binding to the connector : 1/ a TTP conformational change, moving through a domino-effect from the tail tip to the connector (Plisson et al, EMBO J) and 2/ a transmission by the TMP. I totally agree with the present findings that disprove the first mechanistical hypothesis, and place the TMP as the partner of signal transmission. However, TMP is predicted as a long helix, except for two small domains at its N- and C-termini, facing the connector and the baseplate/tail-tip, respectively. Therefore, overall, TMP has no 3D fold, metastable or not. The first trigger occurs when a siphophage bind to its receptor using either its baseplate (saccharidic) or its tail-tip (protein). This may result in a conformational change of the baseplate/tail-tip (observed in the case of some lactococcal phages) providing a mechanical pull to the TMP. Then, as proposed by the authors, the retraction of the TMP would provoke the connector opening. Therefore the TMP transmits a signal from the anti-receptor to the connector. It is not a « trigger » but a « transmitter ».

We agree with the reviewer: we do not think anything definitive can be stated on the TMP fold in the tail assembly other than its ejection being energetically favourable after host receptor binding. We have modified the discussion as follows:

“At the resolution of our EM structures (6 Å), T5 tail tube does not undergo conformational changes upon binding of the tail tip to its receptor. Furthermore, the structure of the building block that forms the different tubes is remarkably conserved in those of siph- and myophage tails, T6SS and R-pyocin, suggesting little or no flexibility of the structured core of the protein. We suggest that in all these structures, the TTP would have only a scaffolding function and would not be involved in signal transduction or in expulsion of the TMP. It is possible that previous contradictory observations on the siphophage SPP1¹⁰ resulted from negative stain artifacts.

Hence, in Siphoviridae, we propose both signal transduction and membrane perforation functions to be fulfilled by the TMP: the TMP would be constrained in a metastable state within the tail assembly, as the sheath of contractile devices^{3,14} and for the Podoviridae core protein³⁵. Interaction of the Receptor Binding Proteins with the host receptor would destabilize this metastable state, leading to expulsion of the TMP from the tail tube and its insertion in the cell wall²⁰. Moreover, the membrane insertion property and muralytic activity described for the C-terminal domains of T5 TMP would serve as complementary forces driving TMP ejection/refolding²⁰. The exit of the TMP at the level of the head-to-tail connector would in turn induce conformational changes in the latter, resulting in its opening fro DNA release³⁶.

In Myoviridae and the other bacterial contractile devices, contraction of the sheath provides the energy to perforate the cell wall^{3,14,37}. It does not however seem to be the signal that opens the capsid in Myoviridae, as contracted tails with full capsids have been isolated³. The signal for capsid opening was proposed to be given by TMP ejection² or conformational changes in the TTP³⁸. Given the remarkable conservation of structures and tail organization (tube, inner baseplate)¹⁵ within Siphoviridae and Myoviridae phages, we propose that the TMP of Siphoviridae and Myoviridae could have a similar role in signal transduction."

Finally, the siphoviridae mechanism for DNA release might be totally different between siphoviridae and myoviridae. In the latter case, the baseplate conformational change activates the sheath contraction and expels the tail tube. The TMP is not involved in this mechanism. To my knowledge, the trigger for TMP release and DNA ejection in Myoviridae is not documented, but may be also linked to the large mechanical stress occurring during tail contraction.

Indeed, as indicated in the discussion, there is no link between sheath contraction and DNA ejection in Myoviridae, and the mechanism of capsid opening has not been investigated. However, given the remarkable structural homology between siphoviridae and myoviridae tails, we propose a similar role of the TMP. A role of the TMP in capsid opening in myoviridae has indeed already been proposed in the literature. We have rephrased our conclusion on that point to soften it (see above).

Reviewer #2 (Remarks to the Author):

Summary.

Arnaud et. al determined the crystal structure of the tail tube protein of phage T5 by X-ray crystallography, analyzed the structure of phage tails with the help of electron microscopy, and performed mutagenesis to address some aspects of tail assembly. The structures of the phage tail in its pre- and post- host cell attachment states have been determined. It showed that the protein occupying the central channel of the tail leaves this channel upon receptor binding. Recombinantly expressed tail protein pb6 assembles into a structure that is virtually identical to that of the post-attachment empty tail. Combining all the observations and taking into account similarities of non-contractile and contractile tails, the authors put forward a hypothesis that the tape measure protein is "stored" in the tail's channel in a metastable state that promotes its expulsion upon interaction of the tail with a receptor on the host cell surface.

Major concerns.

None.

Minor concerns.

The MS will benefit from reading by a colleague who (hopefully) will be able to pick up additional technical and language imperfections besides the ones given below.

The text was proof read by a native English-speaking colleague, so that it has now much improved.

Lines 27-30. These are well studied for the long, contractile tailed Myoviridae¹⁻⁴ and short tailed Podoviridae⁵⁻⁸. For the large family of long flexible tailed Siphoviridae, the

mechanism by which receptor binding, occurring at the tail distal end, promotes DNA ejection remains unclear.

To the contrary, people working on Myo- and Podophages believe that “the mechanism, which receptor binding, occurring at the tail distal end, promotes DNA ejection” is much much clearer in Siphophages considering that NOT A SINGLE INSTANCE of tail tube receptor is known for Myophages or Gram-negative Podophages.

Indeed. We had shortened the introduction in the first version of this paper due to size limit. Because this is not the case any longer, we have elaborated a bit more:

“Phage tail architectures and strategies of cell wall recognition and perforation are different for each family of tailed phages: Myoviridae use a “syringe-like” mechanism, whereby the long and straight contractile tail “drills” the cell wall with a metal-loaded needle¹⁻⁴. For the short tailed Podoviridae, the sequence of events has also been investigated and proteins involved in DNA delivery have been identified⁵⁻⁸. Concerning the large family of the Siphoviridae, the structural information⁹ has yet to detail the mechanism by which receptor binding promotes DNA ejection.”

Line 50. New paragraph.

done

Line 63. Although T5-like TTPs have no homologues in the databases... I do not understand. I thought you already made a pretty convincing case that pb6 is TTP. All TTPs are homologous. Please rephrase or clarify. Did you mean that you cannot see this relationship from the analysis of the amino acid sequence alone? What about HHpred?

Indeed, it is on the sequence level that no homologues were found in the databases, even using the powerful HHPRED software. We have modified the text to make that clearer:

“Although on a sequence level, T5-like TTPs have no homologues in the databases according to the PSIBlast and HHPred software tools, the structure of each subdomain reveals structural homology with TTPs of other sipho- and myophage, distal tail proteins of siphophages, T6SS tube proteins, the tube protein of R-pyocin and baseplate hub proteins from myophages and T6SS (Fig. 1c, Extended Data Fig. 1b and Table 2).”

Line 76-79. Electron density is present in the 40-Å wide lumen of the tube, due to the presence of the TMP pb2. However, no structural information can be derived from this density as three-fold symmetry and averaging along the tube will have blurred it. And _helical_ symmetry?

As described in the MatMet section, no helical symmetry was applied, as the pitch of the boxing along the tail tube was smaller than that of the tube. It was checked that indeed helical symmetry did not improve the reconstruction. See also response to point 3 of reviewer 3.

Lines 69-70. Such a duplication has been observed before in myophage and T6SS hub proteins. The correct reference here is <http://www.nature.com/nature/journal/v415/n6871/abs/415553a.html> This have been known for about 10 years prior to ref. 12...

Thank you for pointing this out. We have added the original reference, however keeping also ref 12, which shows the structural homology between the duplicated domain and the Hcp fold.

As you cannot be sure what you actually see inside the channel, you have to be more careful in interpreting this density: Electron density is present in the 40-Å wide lumen of the tube, which most likely corresponds to TMP pb2.

Indeed, we cannot be sure this density corresponds to pb2. The text was corrected as suggested.

Line 80. Chimera correlation 0.89

This does not make sense for people not familiar with UCSF Chimera (reference needed anyway). If you give the correlation coefficient here, you have to give additional parameters - the radius around the atoms, or perhaps, how many atoms (out of all) are *in the density*).

Details have been added as requested:

“The X-ray structure of pb6 monomer fits remarkably well in the tube density (Chimera²⁷ correlation 0.89 using a pb6 map simulated from atoms at a resolution of 6Å, 879/6657 atoms being outside the contour), except for the N-terminus, loop β13-β14, loop β17-β18 and helix α5 positioned at interfaces between monomers (Fig. 3a).”

Line 82. Remarkably, these elements

“Remarkably” is a wrong word. Besides, you have remarkably two lines above (line 80). Interestingly will be more appropriate here.

Corrected

Lines 95-97. However, the twist of the tube differs, which could stem from the long loops being longer in pb6 (Extended Data Fig. 5d). So, what are the helical parameters and how they compare with those of the pyocin/T4 tube?

The twist of the pyocin and T4 tube have been added for a comparison:

“However, the twists of the tubes differs (39.1° vs. 18.3° x 2 = 36.6° for pyocin and 17.9° x 2 = 36.8° for T4, Extended Data Fig. 5d), which could stem from the long loops being longer in pb6.”

Line 99. New paragraph.

done

Extended Data Table 3: Summary of the performed mutants. p. indicates punctual protein mutations.

You probably meant “point mutations” not “punctual”, correct?

Indeed, now corrected

Reviewer #3 (Remarks to the Author):

In this study by C-A Arnaud and colleagues, the authors have used X-ray crystallography and TEM to study the bacteriophage T5 tail tube structure with a goal to elucidate the mechanism behind DNA ejection. The authors solved the 2.2Å resolution structure of the TTP and the empty and FhuA (host receptor)-interacted tail tube at 6-9Å resolution. This is a robust piece of work and could be published subject to attending to the comments/criticisms and questions provided below.

Main text

1. The author's argument that the empty and FhuA-including tubes are effectively identical is constrained by the low 8Å resolution reconstructions. Is that resolution sufficient to claim that there is indeed no conformational change and that mere interaction of the tube tip to host receptor instigates signal transduction? This referee strongly feels that a higher resolution (e.g. near atomic resolution) and/or other complementary biochemical/mutational studies is needed and the authors should soften their stance in the title, abstract and the last paragraph – a central theme of their study and the phrase “.elucidates trigger..” in title is clearly overambitious.

This issue has also been raised by referee #1. The title, abstract and discussion have been changed to soften the conclusion. 6Å resolution is not atomic resolution, and we cannot exclude small changes, not visible at this resolution.

2. Are there other examples tail tube formed by trimers. Indeed the two pseudo similar subdomains in pb6 structure effectively leads to a traditional hexameric tube protein assembly as in many other tail tubes.

To our knowledge, there is one other example of a phage tail having a three fold symmetry, that of ΦCbK, as mentioned in the original paper (Effantin et al, JMB, 2006). We have added these references in the text:

“Phage T5 is a Siphoviridae infecting the Gram-negative host Escherichia coli. Its overall structure has been determined¹⁶, showing an unusual three-fold symmetry of the tail tube. To our knowledge, such a three-fold symmetry in a tail tube has been observed in only one other case, for siphophage ΦCbK^{17,18}.”

3. Could the author further comment why the loops at the N-terminus are essential for tube formation even though they find these loops to be disordered in the crystal structure?

The crystal structure is that of the monomer, which needs not have the same structure than pb6 within the fibres, as has already been seen in other phage/T6SS proteins. In particular, the structure of Hcp of different bacteria has been determined. The protein always crystallises as a hexamer. Depending on the crystal form however, the long loop is either resolved (PDB codes 3V4H, 1Y12, 3EAA, 2WX6) or not (PDB codes 4TV4, 3HE1, 4HKH). It is clear from the available structure of the two tubes (that of T4 and of pyocin), that this long loop glues the different rings together within the tube, i.e. it is not necessary for the stability of the isolated monomer. We have been more precise in the text:

“We have determined the structure of pb6 monomer at 2.2 Å resolution (Extended Data Table 1).”

And later in the text:

“This is a common phenomenon among phage proteins, where oligomerisation of a protein is regulated by interaction with its partner, and that unstructured regions play a central role in regulating assembly³⁰.”

Material and Methods

1. Line 165: After 1 h and 40 minutes...

Corrected

2. Line 234 – this first sentence is unclear. What does 6 repeats stand for?

*The T5 tail is very flexible. In order to get a straight segment of tail, we boxed out part of a tail. The height of this boxed segment corresponds to ~6 trimeric stacks of the tail (6 * 40.6 Å).*

How exactly was the back projection carried out?

The segment was verticalised and back projected in SPIDER assuming no out of plane and a 3-fold symmetry.

Overall why was a initial model based on negative stain analysis necessary to include as input in RELION.

The initial model was calculated from cryo-EM images (not negative stain). Then a larger dataset was analysed in Relion using this initial model. To make that clear, the first paragraph of the “3D reconstruction” methods was modified as follows:

“An initial model was created by back-projecting a single straight T5 tail (total height of ~6 trimeric stacks) and by applying 3-fold symmetry using SPIDER³⁰. This initial model was refined using a classical projection matching method as described³¹. Briefly, the Contrast Transfer Function (CTF) corrected tail images were boxed into 128 x 128 pixel windows with an overlap of $\frac{3}{4}$ of the box. The generated stacked images (1000 images) were aligned against the projections of the model and averaged into classes. The classes were back-projected to obtain a new 3-fold symmetrised model. After 10 cycles of refinement, we obtained a map at ~10 Å resolution in which the Xray structure of pb6 could be fitted without ambiguity. In order to improve further the resolution, a larger dataset was processed as described below.”

Did the author try a featureless hollow cylinder as a starting model to check any model bias?

No we had not. The helical and 3-fold symmetry of T5 tail has been determined in two previous studies (Guenebaut, V., Maaloum, M., Bonhivers, M., Wepf, R., Leonard, K. & Horber, J. K. (1997). TEM moire patterns explain STM images of bacteriophage T5 tails. Ultramicroscopy, 69, 129–137 and Effantin, G., Boulanger, P., Neumann, E., Letellier, L. & Conway, J. F. Bacteriophage T5 structure reveals similarities with HK97 and T4 suggesting evolutionary relationships. J. Mol. Biol. 361, 993–1002 (2006)). The initial model we obtained with SPIDER was at ~10 Å resolution and the Xray structure fits very well into the density. It therefore validates the correctness of the initial model in our opinion.

Nevertheless, we did the image analysis with a featureless hollow cylinder as a starting model and we obtained the same result as with our previous approach.

The following sentence was modified in the text:

"The low resolution 3D model (see above) or a featureless cylinder were used as initial references for a first 3D refinement imposing C3 symmetry.

3. The authors state that setting the separation of overlapping boxes to ~40Å (axial rise) will allow all helix views to be available. How can the authors be sure of that? Why could not they use IHRSR or SPRING to ensure helix symmetry is applied completely.

At the time the reconstructions were done, helical symmetry was not implemented in RELION. We therefore choose an overlap between successive segments as large as possible so that nearly all helix views are included. As the reviewer mentioned, we were also concerned by an improper helical symmetry application. Thus, we did another refinement of the map in RELION by running at the end of each RELION iteration the two programs hsearch and himpose from the IHRSR package on the two half maps. This "hybrid" refinement lead to a 3D map almost identical to the one obtained with only 3-fold symmetry imposed. Later, when RELION 2 (which handles helical symmetry) was released, we again tested our dataset with this new software and obtained a similar result. Overall, by setting the separation of overlapping boxes close to the axial rise, we think we include almost all helix views so that imposing helical symmetry does not bring anything more.

The following sentence was added:

"This was confirmed later by two different refinement strategies, both enforcing helical symmetry, using IHRSR (The iterative helical real space reconstruction method: surmounting the problems posed by real polymers. Egelman EH. J Struct Biol. 2007 Jan;157(1):83-94) and RELION 2 (D. Kimanius, B.O. Forsberg, S.H.W. & E. Lindahl (2016) "Accelerated cryo-EM structure determination with parallelisation using GPUs in RELION-2" eLife, e18722). Both approaches gave 3D density maps nearly identical to the one obtained without enforcing helical symmetry (data not shown).

4. The figures are uniformly of poor quality and should be redone. For instance Extended Data Figure 6 is particularly poor as the different colours are barely distinguishable

Very different colours were chosen for the different maps in Figure 6, but this figure is difficult to do, for the maps nearly completely overlap. To improve comprehension, fewer but thicker sections, displayed larger, are shown in the new figure, and films showing the overlay are proposed in Supplementary information. The other figures were also modified to improve quality and comprehension.

In the view shown in Fig. 1a Nt end is not visible.

The orientation of the molecule has been slightly changed so that it is now visible.

Fig.2 should be recast with panel numbers 1,2, 3,and 4 – the current assignment mentioned in the legend is confusing.

All panels have now been labelled to improve comprehension.

There are too many bad grammar and peculiar English usages. It is mandatory to have their manuscript properly proof read

The text was proof read by a native English-speaking colleague, so that it has now much improved. As for the English usages, it is stated on Nature Communication web site that the journal uses Oxford English Spelling (<https://www.nature.com/ncomms/submit/how-to-submit>). We have thus kept the English spelling (colour, symmetrised, etc)

(egregious errors e.g. what is peptidic in line 282, PDB not PBD in Extended Data Table 2).

These errors have been corrected

REVIEWERS' COMMENTS:

Reviewer #1 (Remarks to the Author):

The authors have answered to the requests and modified the text in several part in a satisfying way. The figures have been tremendously improved. All in all, this results in a excellent paper, accessible to a general scientific public. I am totally favourable to the publication of the manuscript in its revised form.

Reviewer #3 (Remarks to the Author):

By and large, the authors have responded to the concerns of this and the other two referees so that the manuscript is improved. Some issues that I feel still need to be attended to, are provided below. The issue of language remains troubling – and I have tried to alter some sentences so as to be both scientifically and grammatically correct. I would like to stress that correct language usage is not just a cosmetic issue, it is critical to convey to the reader results appropriately – see below for instance. Given that, the revised version still needs correction of irritating mistakes (e.g. in line 204, see below), I am not sure how many errors still persist!!

Suggested changes in abstract:

Line 17:.....R-pyocin) that form hexameric rings. Structures....

Line 18were determined by ...

Line 19:"resolution. Structure comparisons (please state with what?) reveals....

Line 20:.....capsid through conformational

Main text alterations for clarity

Line 167: Negative stain artifacts – without any evidence this sentence is inappropriate.

Revise as ...resulted from differences in specimen preparation – negative staining versus frozen-hydrated.

Line 168-171: Hence, In Siphoviiriade, we propose that bothperforation functions are fulfilled by the TMP with the TMP constrained... assembly, as for the sheath of ...

Line 172: host receptor serving to destabilize this

Line 181: ... was proposed to be generated by TMP...

Line 184:.....and Myoviridae may have ...

Material and Methods:

Line 204: Replace "wheal" (?????) by wheel

Line 274:straight T5 tail (total of ~6 consecutive trimeric stacks)...

Legends to figures:

Figure 3; Line 513: Unattributed densities are indicated by an asterisk-- what are these densities possibly due to -the authors should comment e.g. any clues from flexible fitting, or is it that the ring structure in the tube is indeed quite different from the x-ray monomer structure.

Figure 4a Most of it is over saturated and this image needs to be recast

REVIEWERS' COMMENTS:

Reviewer #1 (Remarks to the Author):

The authors have answered to the requests and modified the text in several part in a satisfying way. The figures have been tremendously improved. All in all, this results in a excellent paper, accessible to a general scientific public. I am totally favourable to the publication of the manuscript in its revised form.

> We thank the reviewer for his enthusiasm and positive feedback on our work!

Reviewer #3 (Remarks to the Author):

By and large, the authors have responded to the concerns of this and the other two referees so that the manuscript is improved. Some issues that I feel still need to be attended to, are provided below. The issue of language remains troubling – and I have tried to alter some sentences so as to be both scientifically and grammatically correct. I would like to stress that correct language usage is not just a cosmetic issue, it is critical to convey to the reader results appropriately – see below for instance. Given that, the revised version still needs correction of irritating mistakes (e.g. in line 204, see below), I am not sure how many errors still persist!!

We thank reviewer #3 for pointing out some mistakes and mis-spelling. Typos were unfortunately still present, but the text was again carefully proof read to hopefully remove them all.

Suggested changes in abstract:

Line 17:.....R-pyocin) that form hexameric rings. Structures....

> This adds too many words.

Line 18were determined by ...

> changed as suggested

Line 19:"resolution. Structure comparisons (please state with what?) reveals....

> changed so as to add as little words as possible.

Line 20:.....capsid through conformational

> changed as suggested

Main text alterations for clarity

Line 167: Negative stain artifacts – without any evidence this sentence is inappropriate. Revise as ...resulted from differences in specimen preparation – negative staining versus frozen-hydrated.

> The sentence was complemented to clarify our statement.

Line 168-171: Hence, In Siphoviiriade, we propose that bothperforation functions are fulfilled by the TMP with the TMP constrained...

> changed

assembly, as for the sheath of ...

> changed as suggested

Line 172: host receptor serving to destabilize this

> changed

Line 181: ... was proposed to be generated by TMP...

> *changed as suggested*

Line 184:.....and Myoviridae may have ...

> *changed*

Material and Methods:

Line 204: Replace “wheal” (?????) by wheel

> *changed as suggested*

Line 274:straight T5 tail (total of ~6 consecutive trimeric stacks)...

> *changed as suggested*

Legends to figures:

Figure 3; Line 513: Unattributed densities are indicated by an asterisk-- what are these densities possibly due to -the authors should comment e.g. any clues from flexible fitting, or is it that the ring structure in the tube is indeed quite different from the x-ray monomer structure.

> *It is very clearly stated in the text (lines 108-111) and shown in Supplementary Figure 5 that these extra densities could be fitted with flexible elements of the crystal structure of the protein. This is also mentioned in the legend of Figure 3: “Side view of the pseudo-atomic model of the tail tube after flexible fitting of pb6 into each segment (see also Supplementary Fig. 5).”*

Figure 4a Most of it is over saturated and this image needs to be recast

> *Indeed. The image has been changed to a clearer one.*